# Anti-*Haemophilus* Activity of Selected Essential Oils Detected by TLC-Direct Bioautography and Biofilm Inhibition

**DOI:** 10.3390/molecules24183301

**Published:** 2019-09-11

**Authors:** Viktória Lilla Balázs, Barbara Horváth, Erika Kerekes, Kamilla Ács, Béla Kocsis, Adorján Varga, Andrea Böszörményi, Dávid U. Nagy, Judit Krisch, Aleksandar Széchenyi, Györgyi Horváth

**Affiliations:** 1Department of Pharmacognosy, Faculty of Pharmacy, University of Pécs, H-7624 Pécs, Hungary; balazsviktorialilla@gmail.com (V.L.B.);; 2Institute of Pharmaceutical Technology and Biopharmacy, Faculty of Pharmacy, University of Pécs, H-7624 Pécs, Hungary; bai0311@gmail.com (B.H.); szealex@gamma.ttk.pte.hu (A.S.); 3Department of Microbiology, Faculty of Science and Informatics, University of Szeged, H-6726 Szeged, Hungary; kerekeserika88@gmail.com; 4Department of Medical Microbiology and Immunology, Medical School, University of Pécs, H-7624 Pécs, Hungary; kocsis.bela@pte.hu (B.K.); adorjanvarga@ymail.com (A.V.); 5Institute of Pharmacognosy, Faculty of Pharmacy, Semmelweis University, H-1085 Budapest, Hungary; aboszormenyi@gmail.com; 6Department of Genetics and Molecular Biology, Institute of Biology, Faculty of Sciences, University of Pécs, H-7624 Pécs, Hungary; davenagy9@gmail.com; 7Department of Food Engineering, Faculty of Engineering, University of Szeged, H-6724 Szeged, Hungary; krisch@mk.u-szeged.hu

**Keywords:** essential oil, clove, thyme, cinnamon bark, peppermint, anti-biofilm activity, Pickering nano-emulsion, *Haemophilus influenzae*, *Haemophilus parainfluenzae*

## Abstract

Essential oils (EOs) are becoming increasingly popular in medical applications because of their antimicrobial effect. Direct bioautography (DB) combined with thin layer chromatography (TLC) is a screening method for the detection of antimicrobial compounds in plant extracts, for example, in EOs. Due to their lipophilic character, the common microbiological assays (etc. disk diffusion) could not provide reliable results. The aim of this study was the evaluation of antibacterial and anti-biofilm properties of the EO of cinnamon bark, clove, peppermint, thyme, and their main components against *Haemophilus influenzae* and *H. parainfluenzae*. Oil in water (O/W) type Pickering nano-emulsions stabilized with silica nanoparticles from each oil were prepared to increase their water-solubility. Samples with Tween80 surfactant and absolute ethanol were also used. Results showed that *H. influenzae* was more sensitive to the EOs than *H. parainfluenzae* (except for cinnamon bark oil). In thin layer chromatography-direct bioautography (TLC-DB) the ethanolic solutions of thyme oil presented the best activity against *H. influenzae*, while cinnamon oil was the most active against *H. parainfluenzae*. Pickering nano-emulsion of cinnamon oil inhibited the biofilm formation of *H. parainfluenzae* (76.35%) more efficiently than samples with Tween80 surfactant or absolute ethanol. In conclusion, Pickering nano-emulsion of EOs could inhibit the biofilm production effectively.

## 1. Introduction

Essential oils (EOs) have been widely used for antimicrobial, medicinal and cosmetic purposes. In the European Union, these plant extracts can be found in foods (as flavorings), perfumes (as fragrances) and pharmaceuticals (as active ingredients) [1,2]. The significance of the EOs and their components as antimicrobial substances are increasing, due to antibiotic-resistant pathogens [3,4]. EOs may represent the richest available reservoir of novel therapeutics [5]. However, the reliability of the common antimicrobial assays used for EOs is questionable because of their non-water soluble property [6].

Direct bioautography (DB) combined with thin layer chromatography (TLC) is a rapid and sensitive screening method for the detection of antimicrobial compounds. Test microorganism is capable of growing directly on the TLC plate, so each step of the assay is performed on the sorbent. Similar to the widely used antimicrobial screening methods (e.g., broth macro- and microdilution), thin layer chromatography-direct bioautography (TLC-DB) should be carried out under controlled conditions, since the experimental parameters (for example, solvents, sample application, resolution of compounds, type of test microorganism, incubation time) may influence the result [7]. This assay is capable of testing multicomponent and lipophilic extracts, e.g., EOs. The applicability of bioautography to detect antimicrobial compounds effective against plant and human pathogenic bacteria has been reported in the literature [8,9,10]. However, there is only a few studies in which respiratory pathogens were included in TLC-DB method [11,12]. According to the data of the World Health Organization (WHO), lower respiratory tract infections are responsible for 5% (3.1 million people) of deaths worldwide [13]. EOs offer effective treatment in the respiratory tract infections because of their volatility and antibacterial effect. We tested other respiratory tract pathogens, such as *Streptococcus* species and *Pseudomonas aeruginosa*, too [14,15]. The mode of action of EOs is not fully understood, but the prevention of the bacterial biofilm formation may be suggested. Therefore, we decided to study the biofilm inhibition potential of our EO samples, including respiratory tract pathogens into our experiments.

A biofilm comprises any group of microorganisms in which cells stick to each other and often also to a surface. These adherent cells become embedded within a slimy extracellular matrix that is composed of extracellular polymeric substances (EPS) [16,17]. Biofilms have been found to be involved in a wide variety of microbial infections (e.g., bacterial vaginosis, urinary tract infections, catheter infections, middle-ear infections) in the body, by one estimate 80% of all infections. About 80% of cystic fibrosis patients have a chronic lung infection, caused mainly by *P. aeruginosa* growing in a non-surface attached biofilms [18]. Infections associated with the biofilm growth usually are challenging to eradicate. It is mostly due to the fact that mature biofilms display tolerance towards antibiotics and the immune response [19]. Most of the publications focus on the inhibition of bacterial biofilm produced by foodborne or dental pathogens [20,21]. Therefore, it was worth involving the pathogens of respiratory tract infections in the studies, in which the effect of our EO samples on biofilm formation produced by respiratory tract bacteria was examined.

Therefore, the aim of this study was the evaluation of antibacterial properties of the EO of cinnamon bark (*Cinnamomum verum* J. Presl.), clove (*Syzygium aromaticum* (L.) Merr. And Perry), peppermint (*Mentha* x *piperita* L.), thyme (*Thymus vulgaris* L.), and their main components (trans-cinnamaldehyde, eugenol, menthol, and thymol) against the Gram-negative bacteria, *Haemophilus influenzae* and *H. parainfluenzae* using TLC-DB. Furthermore, the biofilm inhibition of different formulation of our EO samples was also performed. The chemical composition of the EOs was measured by gas-chromatography-mass spectrometry (GC-MS).

## 2. Results

### 2.1. Chemical Composition of EOs

Chemical analyses of EOs were performed by GC-FID and GC-MS techniques. Identified compounds and percentage evaluation of the oils are shown in Table 1.

Eugenol (78.8%) was the main component in the EO of clove. Cinnamaldehyde (63.7%) was the main component in the oil of cinnamon bark. Menthol (50.4%) was the characteristic compound in the peppermint EO. In the thyme oil thymol (39.8%) was identified as the main constituent.

### 2.2. TLC-DB

#### 2.2.1. Antibacterial Activity of EOs

In the TLC-DB method, the activity of the EOs without and with separation was tested against *H. influenzae* and *H. parainfluenzae.* In the case of activity of the EOs without separation, the development with mobile phase was not prepared; therefore, the activity of the “total” extract (EO) was examined [22]. Figure 1 shows the activity of the EOs without TLC separation. The diameter of the inhibition zones was expressed in cm. From the stock solution of EOs 1 µL was applied (equivalent to 0.2 mg undiluted EO) on the TLC plate. The *H. influenzae* was more sensitive to the EOs than *H. parainfluenzae* (except for cinnamon bark oil). Absolute ethanol as negative control did not inhibit the growth of both bacteria. The 0.2 µL solution of the antibiotic sample (amikacin, equivalent to 0.05 mg antibiotic) was effective against both *Haemophilus* strains. Ethanolic solutions of thyme oil presented the best activity in case of *H. influenzae*, while cinnamon bark oil was the most potent against *H. parainfluenzae*. Peppermint showed moderate activity in case of both pathogens (0.51 cm against *H. influenzae* and 0.31 cm against *H. parainfluenzae*). In our TLC-DB assay, the tested EO samples did not show more effective activity than the positive control, but their combination (antibiotic and EO) might be the aims of the following assays.

#### 2.2.2. Antibacterial Activity of Main Components of EOs by TLC-DB Method

Generally, the antibacterial activity of the EOs seems to be associated with their most abundant compounds, but the effect of the minor compounds should also be taken into consideration. In the oil of cinnamon bark, cinnamic aldehyde and eugenol components, as well as their standards, showed activity in the case of both bacteria (Figure 2). Moreover, α-terpineol (Rf = 0.35) in the EO of cinnamon showed activity against both bacteria. A-Terpineol was identified according to the GC-MS result and Wagner and Bladt [23]. Eugenol, as the main compound of the clove oil (Rf = 0.52), was active against both *Haemophilus* strains (Figure 2). In the peppermint oil, several compounds had antibacterial activity at the tested concentration. Menthol (Rf = 0.31) in the peppermint oil and the standard of menthol inhibited the growth of bacteria (Figure 2). Other active compounds of peppermint oil include 1,8-cineole, isomenthon, menthon, and isomenthyl acetate (according to GC-MS and Wagner and Bladt [23]. In the oil of thyme, thymol-carvacrol and the standard, thymol had antibacterial activity (Figure 2). At Rf = 0.33, linalool was identified as an active compound according to GC-MS result and Wagner and Bladt [23].

### 2.3. Preparation and Characterization of Pickering Nano-Emulsions of EOs

The preparation of the stable Pickering nano-emulsions has been described before [24]. We had considered the emulsion to be stable when the droplet size did not change for at least 24 h; creaming, sedimentation and disproportionation did not occur. The droplet size and stability of Pickering and conventional emulsions of EOs can be seen in Table 2. The Pickering emulsions are more stable than conventional emulsions, the difference in stability is most obvious in a case of peppermint EO; its Pickering emulsion form remains stable for at least five months, while its conventional emulsion is stable for only one month. In the case of clove EO, there is no difference in stability between the Pickering and conventional emulsions; they remained stable for only two weeks. Conventional emulsions are stabilized with Tween 80, while Pickering nano-emulsions were stabilized with silica nanoparticles with a mean size of 20 nm and surface modified with ethyl groups (20ET).

### 2.4. Anti-Biofilm Activity

We examined the inhibitory effect of the EOs (half of the MIC concentration) according to a previous study [25]. Three different formulations of the EOs were tested. The anti-biofilm formation activity of the EOs was calculated and demonstrated in the term of inhibitory rate according to the following equation: Inhibitory rate = (1−S/C) × 100% (C and S were defined as the average absorbance of control and sample groups respectively) [26]. Our results showed that not only the EOs samples with Tween80 surfactant had an antibacterial effect, but also the EO samples with absolute ethanol and their Pickering nano-emulsion forms inhibited the biofilm formation. Among the controls, Pickering nano-emulsion without EO showed the lowest activity (1.49% inhibitory rate) (Figure 3 and Figure 4). It should be highlighted that the Pickering nano-emulsions were the most effective form of EOs against biofilms. The Pickering nano-emulsion of thyme oil showed the highest inhibitory rate (73.64%) against *H. influenzae*. Cinnamon oil in Pickering nano-emulsion form inhibited the biofilm formation of *H. parainfluenzae* (76.35%) most effectively. Among the different formulations, the EO samples with Tween80 showed the lowest activity against biofilm formation. Moreover, in the case of both bacteria, the peppermint oil was the least effective EO among the investigated oil samples. The results of biofilm inhibition assay were in harmony with the results of TLC-DB assay, thyme and cinnamon oils were the most effective among the investigated oils, besides clove also showed potent activity against these respiratory bacteria.

## 3. Discussion

Plants produced a wide variety of secondary metabolites that exhibited antimicrobial activity against a variety of pathogens (bacteria, fungi, and viruses) [27,28,29]. Several suggestions, (or hypothesis) can be found in the literature about their antifungal, and antibacterial mode of action, but some of them need clarification. In this study, the antibacterial and anti-biofilm effects of clove, cinnamon, thyme, and peppermint oils were investigated against *H. influenzae* and *H. parainfluenzae*. The most accepted mechanism of EOs revealed that they could disrupt cell wall and cytoplasmic membrane, leading to lysis and leakage of intracellular compounds [12,30,31,32]. Many bacteria disclosed a high sensitivity to EOs, especially *H. influenzae*, *Stenotrophomonas maltophilia*, *Streptococcus pneumoniae*, *S. pyogenes*, and *S. agalactiae*. Cinnamon, thyme and clove oils showed the strongest inhibitory activity against several bacteria, including *H. influenzae*, *S. pyogenes* and *S. agalactiae*, even against multi-resistant strains [11]. However, the common antibacterial assays (e.g., disk diffusion) are not an appropriate method for non-water soluble extracts and compounds. Therefore, in this study, we focused on the antibacterial potential and anti-biofilm activity of clove, cinnamon, thyme and peppermint oils against *H. influenzae* (DSM 4690) and *H. parainfluenzae* (DSM 8978) using TLC-DB and biofilm inhibitory assays.

TLC-DB is a directly combined application of an analytical method with an in situ bioassay that allows rapid identification of the active compound or compounds in a complex mixture. To the best of our knowledge, we optimized this technique first using *Haemophilus* species. The TLC-DB was optimized for two *Haemophilus* species, but it is necessary to note that attention should be paid to the parameters (e.g., incubation time, the composition of agar for growing the bacterium, etc.) of TLC-DB, which was also confirmed in a previous study [7].

Fabio et al. described that *Haemophilus* species were sensitive to EOs in the following order: Thyme, cinnamon, clove, eucalyptus, sage, and lavender. We determined the highest activity of the thyme, cinnamon, and clove against *H. influenzae* and *H. parainfluenzae*, which was in parallel with the previous observations [12,14].

Our findings showed similar results with Houdkova and co-workers’s [33] and Inouye and co-workers’ [12] results regarding the activity of menthol, menthone and their derivatives, because these EO components are highly responsible for the anti-*Haemophilus* activity. In our research, TLC-DB was optimized with *Haemophilus* species, which is an appropriate assay to detect the antimicrobial activity of EO main compounds. We demonstrated the anti-*Haemophilus* effect of cinnamaldehyde, thymol, menthol, and eugenol. Moreover, some minor components (menthone, isomenthyl acetate, 1,8-cineole, α-terpineole, and linalool) contributed to the antibacterial activity.

In the last decade, the role of natural products derived from medicinal plants for interfering pathogenic biofilms has gained increased attention by the researches [34,35,36,37,38,39,40]. The individual components of the EOs clearly had antibacterial properties, although the mechanism is poorly understood. Therefore, the effect of EOs used in our study on biofilm formation of the two *Haemophilus* species was also investigated. Some previous studies from the literature have already described that cinnamon oil inhibited the biofilm of following pathogens as well, *Campylobacter jejuni, Enterobacter aerogenes*, *E. coli*, *L. monocytogenes*, *P. aeruginosa*, *Salmonella enteritidis*, and *S. aureus* [41,42,43,44] reported the inhibitory effect of menthol, menthone, pulegone, 1,8-cineole, terpinen-4-ol against the biofilm formation of *Salmonella typhimurium*, *E. coli*, *Micrococcus luteus*, *S. aureus*. Clove and thyme oil also showed strong anti-biofilm activity against several Gram-positive (e.g., *S. aureus*, *Brochothrix thermosphacta*, *Lactobacillus rhamnosus*, *L. monocytogenes*, *B. subtilis*, *L. innocua*) and Gram-negative (e.g., *C. jejuni*, *E. aerogenes*, *E. coli*, *P. fluorescens*, *S. enteritidis*) pathogens as well [45,46,47,48,49,50]. The biofilm inhibition of *Haemophilus* species has been only screened with synthetic products: Cefotaxime [51], 1,2,4-triazole-ciprofloxacin [52], garenoxacin [53]. The effect of cinnamon, thyme, clove and peppermint oil against *Haemophilus* species has not been tested earlier.

In the pharmaceutical technology, the formulation of the products and the water solubility of the active ingredients are highly important. In our previous experiment, three different formulations of EOs were prepared used absolute ethanol, Tween80, and Pickering nano-emulsion. Among the EOs, thyme and cinnamon produced the highest inhibitory rates and their Pickering nano-emulsions were the most effective formula [24]. This study showed that the nanotechnological formulated samples had pronounced anti-biofilm effect compared to the non-formulated EOs samples. The samples with absolute ethanol and Tween80 surfactant resulted that the biomass of *Haemophilus* biofilm decreased by half, but using the Pickering-emulsions the biomass of biofilm decreased to one third. Using Tween80 surfactant, it is discernible the decrease of biomass compared to the BHI control. In this case, we cannot exclude the antibacterial effect of the surfactant. The absolute ethanol itself has not strong antibacterial effect, but the EO samples with absolute ethanol resulted in the least biomass reduction.

## 4. Materials and Methods

### 4.1. Essential Oils and Their Components

The EO of clove (Batch number: H7352/1602), cinnamon bark (Batch number: I3201/1609), peppermint (Batch number: H7101/1601), and thyme (Batch number: H3981/1509) were obtained from a Hungarian company (AROMAX Zrt., Budapest, Hungary). Their chemical composition was determined by GC-MS. The main components of the EOs (eugenol, *trans*-cinnamaldehyde, menthol and thymol) were bought from Sigma-Aldrich (Budapest, Hungary).

### 4.2. GC-FID and GC-MS

One microliter of EO samples, diluted in ethanol (10 µL/mL), was injected in split mode, the injector temperature was 250 °C, and the split ratio was 1:50. The analyses were carried out with an Agilent 6890N/5973N GC-MSD (Santa Clara, CA, USA) system equipped with an Agilent SLB-5MS capillary column (30 m × 250 µm × 0.25 µm). The GC oven temperature was increased at a rate of 8 °C/min from 60 °C (3 min isothermal) to 250 °C, with a final isotherm at 250 °C for 1 min. High purity helium was used as carrier gas at 1.0 mL/min (37 cm/s) in constant flow mode. The mass selective detector (MSD) was equipped with a quadrupole mass analyser and was operated in electron ionization mode at 70 eV in full scan mode (41–500 amu at 3.2 scan/s). The data were evaluated using MSD ChemStation D.02.00.275 software (Agilent). The identification of the compounds was carried out by comparing retention times, linear retention indexes, and recorded spectra with the data of authentic standards, and the NIST 2.0 library was also used. The GC-FID were made using a Fisons GC 8000 gas chromatograph (Carlo Erba, Italy). An Rt-β-DEXm (Restek) capillary column, 30 m × 0.25 mm i.d., 0.25 μm film thickness, was used. The carrier gas was nitrogen at 6.8 mL/min flow rate. A 0.2 mL volume of a 0.1% solution of the oil was injected (1 mL EO in 1 mL chloroform). The splitless injection was carried out. The temperatures of the injector and detector were 210 °C and 240 °C, respectively. The oven temperature was increased at a rate of 8 °C/min from 60 °C to 230 °C, with a final isotherm at 230 °C for 5 min. Identification of peaks was made by retention data compared with data obtained by GC-MS and data of standards (Fluka Analytical and Sigma-Aldrich); percentage evaluation was carried out by area normalization. Three parallel measurements were made; RSD percentages were below 4.5%.

To identify the microbiologically active compounds in the separated EOs during TLC-DB, 150 µL of EO was applied onto the TLC layer as 150 mm band, and after the development, the zones of active compounds were scraped off and eluted with 0.5 mL of ethanol for GC-MS.

### 4.3. TLC-DB

#### 4.3.1. Cultivation of Test Bacteria for Dipping

The antibacterial effect of EOs and their main components was screened on *Haemophilus influenzae* (DSM 4690) and *H. parainfluenzae* (DSM 8978) in the laboratory of the Department of Medical Microbiology and Immunology (Medical School, University of Pécs, Pécs, Hungary). For bioautographic assay, bacteria were grown in 100 mL Brain Heart Infusion Broth (BHI) (Sigma Aldrich Ltd., Darmstadt, Germany) with 1 mL supplement B (Diagon Kft., Budapest, Hungary) and 15 µg/mL NAD solution (1 mg/mL) at 37 °C in a shaker incubator at a speed of 60 rpm for 24 h [54]. The bacterial suspension was diluted with fresh nutrient broth to an OD_600_ of 0.4, which corresponds to approximately 4 × 10^7^ colony-forming units (cfu) mL.

#### 4.3.2. Layer Chromatography

We investigated the antibacterial effect of EOs without TLC separation and the antibacterial effect of their components after TLC separation [22]. Chromatography was performed on 5 × 10 cm silica gel 60 F_254_ aluminum sheet TLC plates (Merck, Darmstadt, Germany). EOs were dissolved in absolute ethanol to give solutions containing 100 μL oil in 500 μL absolute ethanol, and 1.0 μL was applied to the TLC plate with Finnpipette pipettes (Merck, Darmstadt, Germany). Absolute ethanol was the solvent control and amikacin (Likacin 250 mg/mL, Lisapharma S.p.A.) as a positive control. 0.2 μL from the positive control (amikacin) and 1.0 μL from the negative control (absolute ethanol) have been applied to the TLC plate. After separation with the mobile phase, the antibacterial activity of the main EO components (thymol, menthol, trans-cinnamaldehyde, and eugenol) was also investigated by TLC-DB. The main components were dissolved in absolute ethanol to give solutions containing 20 mg/mL. From the stock solutions, 0.2 μL (0.004 mg) were applied to the plates. From the previously mentioned EO solutions, 1.0 µL was used. The position of the starting line was 1.5 cm from the bottom and 1.5 cm from the left side. The standards were applied to the TLC plates next to the spots of the oils. After sample application, the plates were developed with the previously optimized mobile phase. For the separation of EOs, toluene:ethyl acetate (95:5) and dichloromethane (in the case of cinnamon bark oil) was recommended as the mobile phase [24]. Ascendant development chromatography was used, in a saturated twin trough chamber (Camag, Muttenz, Switzerland). All TLC separations were performed at room temperature (20 °C). After chromatographic separation, the absorbent layers were dried at 90 °C, for 5 min to remove the solvent completely. Ethanolic vanillin–sulfuric acid reagent [24] was used to visualize the separated compounds. Detection of the separated compounds was performed on Rf value and color of the standards. Evaluation of the separated compounds was also performed under UV light at 254 nm. It should be noted that the TLC plates for bioautography were not treated with ethanolic vanillin–sulfuric acid reagent, because this step interferes with the microbiological steps of TLC-DB.

#### 4.3.3. Post-Chromatographic Detection

After layer chromatography, the TLC plates were treated with the suspension of *H. influenzae* and *H. parainfluenzae*, respectively. Layers were dipped into a 100 mL of bacterial suspension to assure a homogenous distribution and adhesion of bacteria onto the surface of the layers. After immersion, the layers were transferred into a low-wall horizontal chamber (chamber dimension: 20 × 14.5 × 5 cm) and incubated for 2 h at 37 °C. Thereafter for visualization of antibacterial spots, TLC plates were immersed into the aqueous solution of 3-(4,5-dimethylthiazol-2-yl)-2,5-diphenyltetrazolium bromide (MTT, 0.05 g/85 mL) (Sigma Aldrich Ltd., Darmstadt, Germany), for 5 s, and then incubated at 37 °C for 24 h. On the TLC plate, metabolically active bacteria convert the tetrazolium salt, MTT, into formazan dye. White spots (as inhibition zones) against the bluish-violet background indicated the lack of dehydrogenase activity, due to the antibacterial activity of the tested EO or their main compounds. The inhibitory zones (expressed in cm) of EOs without separation were measured with Motic Images Plus 2.0 program (ver. 2.0., Motic, Hong Kong, China).

### 4.4. Statistical Analyses

Statistical analyses were made in R, version 3.1.2 [55]. The measured diameters were analyzed with linear model [56] using the function lm. In our model, the explanatory variables (bacterial species, EO and quantity of EOs), treated as fixed factors. No data transformation was made. Checking the need for transformation was based on graphical evaluation, according to Crawley [57]. For pair-wise comparisons, Tukey post-hoc tests were conducted in with multicomp-package [58] to compare the difference among all experimental set-ups.

### 4.5. Biofilm Inhibition Experiments

#### 4.5.1. Broth Macrodilution Test (BDT)

During biofilm inhibition experiments minimum inhibitory concentration/2 (MIC/2) values of the EOs were used. The MICs were determined with broth macrodilution test (BDT) based on Ács et al. [14].

#### 4.5.2. Preparation of Pickering Emulsion of the EOs

Because of the volatility and non-water soluble characters of EOs, we wanted to make the water-soluble formulation of our EO samples, and eliminate such kind of solvents, e.g., dimethyl sulfoxide (DMSO), from the assay which can generally be used in the microbiological assays, but may influence the results.

For the synthesis and surface modification of silica nanoparticles tetraethoxysilane [TEOS], (Alfa Aesar, [Haverhill, MA, USA], purity 98%), ethyltriethoxysilane [ETES] (Alfa Aesar [USA], purity 96%), absolute ethanol (VWR Chemicals [Budapest, Hungary], AnalaR Normapur, purity ≥ 99.8%), 28 *w*/*w*% ammonium solution (VWR Chemicals [Hungary], AnalaR Normapur, analytical reagent) were used. The stabilizing agent of Pickering emulsions was nanoparticle suspensions, of conventional emulsions was Tween^®^80 (Polysorbate80, Acros Organics, Princeton, NJ, USA).

##### Synthesis, Surface Modification and Characterization of Silica Nanoparticles

Synthesis of hydrophilic silica was based on the work of Stöber, Fink and Bohn [59]. Previously, we performed the optimization of size-controlled silica nanoparticle synthesis process and their surface modification with ETES. Furthermore, the nanoparticle characterization was also completed. The details can be read in our paper [25].

##### Preparation and Characterization of O/W (Oil/Water) Type EOs Emulsions

For the preparation of conventional, surfactant stabilized the emulsion, Tween80 non-ionic surfactant was used, because this chemical is widely used in microbiological experiments and protocols, for solubilizing the non-water soluble, lipophilic molecules, e.g., EOs. We have used 20ET nanoparticles [51] as stabilizing agents for preparation of Pickering emulsions. The concentration of emulsion stabilizing agents was 1 mg/mL in all cases. The mixture of EO, Tween80/silica nanoparticle suspension and water is sonicated for 2 min (Bandelin Sonorex RK 52H, Berlin, Germany) in the pre-emulsification process. The final emulsification was performed with UltraTurrax (IKA Werke T-25 Basic, Staufen, Germany) for 2 min at 13,500 rpm. Each sample was made in triplicates. The emulsion droplet size was determined with dynamic light scattering (DLS) using a Malvern Zetasizer Nano S instrument (Malvern Panalytical Ltd., Malvern, United Kingdom). The stability of emulsions was examined with periodical droplet size measurement. The emulsions were stored at room temperature (*t* = 25 °C) in dark bottles. The EO concentrations were the MIC/2 values against *H. influenzae* and *H. parainfluenzae*.

#### 4.5.3. Anti-Biofilm Activity Test

The biofilms were prepared in 96-well microtiter plate. 200 µL of bacterial culture (4 × 10^7^ cells/mL) was added into each well; then, the microtiter plate was incubated at 37 °C for 4 h in order to help the adhesion of the cells. After the incubation time the non-adherent cells were washed with physiological saline solution. The absolute ethanol, Tween 80 surfactant (1%) and Pickering nano-emulsions of the EOs were used for the experiments. In the experiment, untreated samples were applied, when only BHI medium was added to the bacterial culture. As detergent control, we used Tween80 (1%), 20ET Pickering nanoparticles (1 mg/mL), and as solvent control, absolute ethanol was also prepared. After the treatments, the microtiter plate was incubated again at 37 °C for 24 h. Then the adherent cells were fixed with methanol for 15 min. The biofilms were dyed with 0.1% crystal violet solution for 20 min. The redundant dye was removed. 33 *w*/*w*% of acetic acid was added to each well. Then the absorbance was measured at λ = 595 nm with a microtiter plate reader (BMG Labtech SPECTROstar Nano, Budapest, Hungary). All tests were carried out in six times [60].

## 5. Conclusions

Overall, we can say that the nanotechnological formulation of EOs seem to be a promising solution in anti-biofilm tests, because the Pickering-emulsion without EO has not antibacterial effect, but the Pickering-emulsion with EO resulted decreasing of the biofilm biomass. Cinnamon, thyme, and clove oils in Pickering emulsion showed not only anti-*Haemophilus* activity, but inhibited the biofilm formation in contrast with conventional, surfactant stabilized emulsion or absolute ethanol. We suppose that the enhanced biofilm inhibition properties of the Pickering nano-emulsion of EOs may be attributed to the adsorption of silica nanoparticles on the surface and pores of agar membrane or biofilm. It can be concluded that O/W type Pickering nano-emulsions form of thyme, cinnamon, and clove oils provide a new possibility for the application of EOs in pharmaceutical treatment against *Haemophilus influenzae* and *H. parainfluenzae* caused respiratory tract diseases.

Furthermore, we conclude that TLC-DB is an appropriate assay for detecting the anti-*Haemophilus* activity of non-water soluble extracts with complex composition, e.g., EOs. According to our results, the EO of cinnamon bark, thyme, and clove are promising antibacterial agents against *H. influenzae*, and *H. parainfluenzae* and their biofilm inhibitory capacity may be included in the mode of antibacterial action. Their water soluble Pickering nano-emulsions showed the highest inhibitory rate in the anti-biofilm test; therefore, this formulation may be regarded as relevant preparation for further biological experiments, including scanning electron microscopic and cell line studies.

## Figures and Tables

**Figure 1 molecules-24-03301-f001:**
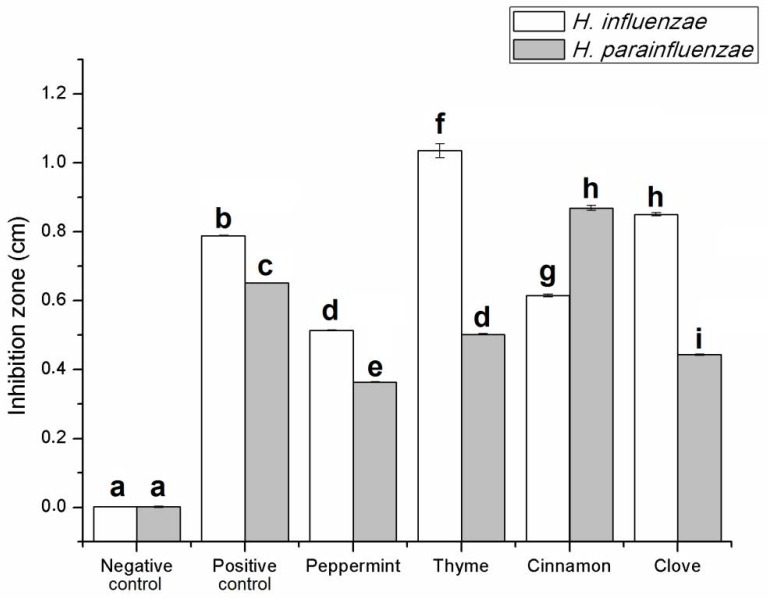
Antibacterial activity of essential oils (EOs) used in this study with direct bioautography (without TLC separation). The diameter of the inhibition zones was expressed in cm. Negative control—absolute ethanol; positive control—amikacin (equivalent to 0.05 mg antibiotic); 1 µL of EO sample (equivalent to 0.2 mg undiluted EO) was applied. Error bars represent S.E.M. Lowercase letters (**a–i**) show pairwise comparison based on Tukey post-hoc test, *p* < 0.05.

**Figure 2 molecules-24-03301-f002:**
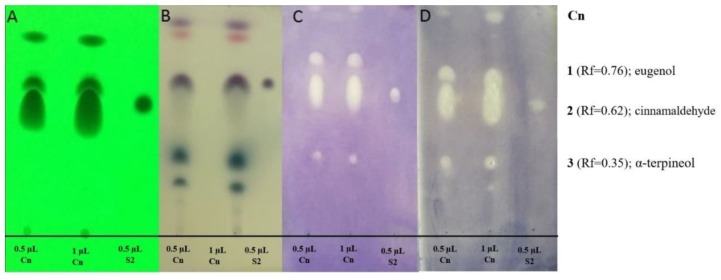
Antibacterial components in the EOs used in this study after TLC-DB. Mobile phases: Dichloromethane (only in case of cinnamon bark oil) and toluene-ethyl acetate 93:7 (*v*/*v*); 0.5 and 1 µL indicated the applied volumes of the EO and the standards. (**A**) TLC plate under UV 254 nm, (**B**) TLC plate after treatment with vanillin-sulfuric acid reagent and documented in visible light, (**C**) TLC-DB assay: Bioautograms using *H. influenzae*, (**D**) TLC-DB assay: Bioautograms using *H. parainfluenzae* (bright zones indicate antibacterial effects); Cn—cinnamon bark oil (200 mg/mL); Cl—clove oil (200 mg/mL); Pp—peppermint oil (200 mg/mL); Th—thyme oil (200 mg/mL); S2—standard of cinnamaldehyde, S3—standard of eugenol, S4—standard of menthol, S5—standard of thymol.

**Figure 3 molecules-24-03301-f003:**
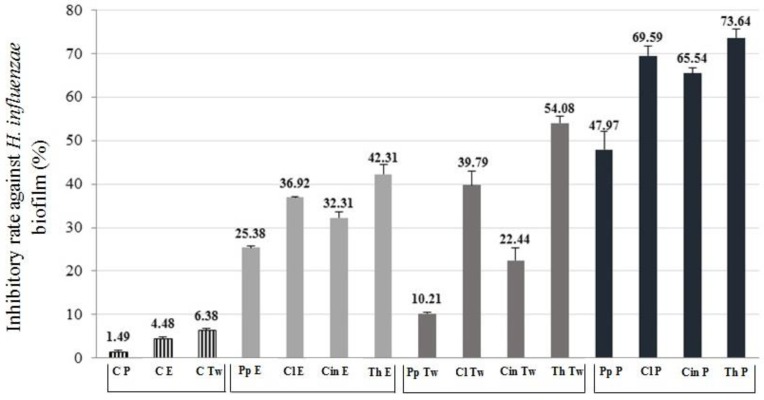
Biofilm inhibition activity of different formulated EOs against *Haemophilus influenzae*. C—control; P—Pickering nano-emulsion form; E—samples with absolute ethanol; Tw—samples with Tween80 surfactant; Pp—peppermint EO; Cl—clove EO; Cin—cinnamon EO; Th—thyme EO. The activity of anti-biofilm formation was calculated and demonstrated in the term of inhibitory rate according to the equation: Inhibitory rate = (1 − S/C) × 100% (C and S were defined as the average absorbance of control and sample groups respectively).

**Figure 4 molecules-24-03301-f004:**
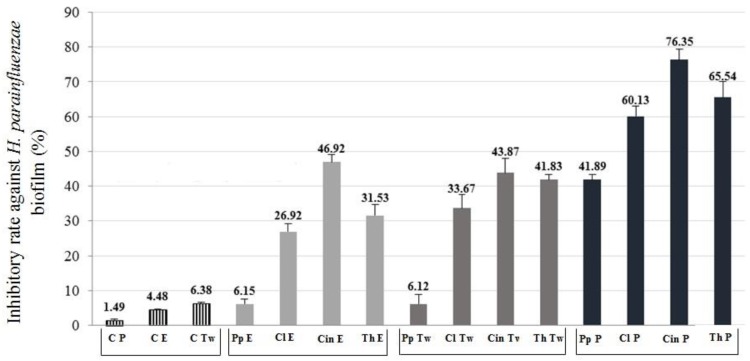
Biofilm inhibition activity of different formulated EOs against *Haemophilus parainfluenzae*. C—control; P—Pickering nano-emulsion form; E—samples with absolute ethanol; Tw—samples with Tween80 surfactant; Pp—peppermint EO; Cl—clove EO; Cin—cinnamon EO; Th—thyme EO. The activity of anti-biofilm formation was calculated and demonstrated in the term of inhibitory rate according to the equation: Inhibitory rate = (1 − S/C) × 100% (C and S were defined as the average absorbance of control and sample groups respectively).

**Table 1 molecules-24-03301-t001:** Average values of volatile compounds from EOs of Peppermint (1), Thyme (2), Clove (3) and Cinnamon (4) from three parallels experiments.

Component	RI	Percentage of Compounds (%)			
		1	2	3	4
α-Pinene	939	1.1	1.0	-	5.1
Camphene	951	-	2.0	-	-
β-Myrcene	992	-	1.0	-	-
α-Terpinene	1017	-	3.2	-	-
*p*-Cymene	1026	-	19.2	-	1.9
Limonene	1044	1.4	-	-	1.8
1,8-Cineole	1046	5.5	4.6	-	2.8
γ-Terpinene	1060	-	6.7	-	-
Linalool	1104	-	5.6	-	4.0
Isopulegol	1150	1.0	-	-	-
Menthone	1156	19.8	-	-	-
Isomenthone	1159	7.0	-	-	-
Menthol	1172	50.4	-	-	-
Isomenthol	1183	4.3	-	-	-
α-Terpineol	1190	-	-	-	2.2
Pulegone	1215	1.9	-	-	-
*trans*-Cinnamaldehyde	1266	-	-	-	63.7
Bornyl acetate	1289	-	1.0	-	-
Thymol	1297	-	39.8	-	-
Isomenthyl acetate	1305	5.5	-	-	-
Eugenol	1373	-	-	78.8	4.6
β-Elemene	1394	-	-	-	-
β-Caryophyllene	1417	1.3	4.2	13.5	4.2
Cinnamyl acetate	1446	-	-	-	9.4
α-Humulene	1452	-	-	4.6	-
β-Cadinene	1473	-	-	1.1	-
Total:	-	99.2	88.3	98.0	99.7

**Table 2 molecules-24-03301-t002:** Properties of Pickering and conventional emulsions of essential oils. Droplet sizes were determined with DLS measurements. Three parallel samples and measurements were made.

Properties of Pickering and Conventional Emulsions of Essential Oils				
Essential Oil	Coil (mg/mL)	Stabilizing Agent	Droplet Size (nm)	Stability
Cinnamon bark EO	0.03	20ET nanoparticles	256.2 ± 12.3	2 months
	0.03	Tween80	274.5 ± 28.5	1 month
Clove EO	0.125	20ET nanoparticles	184.6 ± 8.8	2 weeks
	0.125	Tween80	185.2 ± 10.7	2 weeks
Peppermint EO	0.105	20ET nanoparticles	308.7 ± 15.5	5 months
	0.105	Tween80	248.9 ± 4.	1 months
Thyme EO	0.055	20ET nanoparticles	180.5 ± 6.4	4 months
	0.055	Tween80	163.2 ± 1.3	1 month

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
