# Peer review of "Anti-Haemophilus Activity of Selected Essential Oils Detected by TLC-Direct Bioautography and Biofilm Inhibition"

_molecules, 2019, doi:10.3390/molecules24183301_

Round 1

Reviewer 1 Report

In this manuscript the authors report the antibacterial and anti-biofilm properties of the essential oils of cinnamon bark, clove, peppermint and thyme against Haemophilus influenzae and H. parainfluenzae. Interestingly, they find antibacterial activity of the EOs. More importantly they find that the Pickering nano-emulsion of EOs could inhibit the biofilm production effectively. They also show that TLC-DB is a good method for detecting anti-Haemophilus activity of EOs.

The manuscript is well written. The methodology is appropriate for the study and the findings are clear. However the figures could be improved.

Overall the research is well developed, but I suggest to accept the paper after some revision:

-The following sentences should be corrected:

Line 23 (etc. disc diffusion).

Line 98 Against of both

Line 113 Menthol and of standard

Line 120 We consider the emulsion to stable….

Line 128 which is stable for only 1 months

Remove the paragraph from line 150 to line 152

Line 237 therefore, these studies contribute to the mode of action of EOs that their main compounds….

Line 264 pronounced anti-biofilm effect, than the non-formulated EOs samples, too.

Line 305 “1. Cultivation of test bacteria for dipping” should be “4.3.1. Cultivation of test bacteria for dipping”

Line 314 “2. Layer chromatography” should be “ 4.3.2. Layer chromatography”

Line 380 “1. Synthesis, surface modification……” should be “4.5.2.1. Synthesis, surface modification……”

Line 381 Synthesis of hydrophilic silica was based on ……

Line 386 “2. Preparation and characterization of O/W type …….” should be “4.5.2.2. Preparation and characterization of O/W type …….”

-In Fig. 1 the type of extracts that are used to measure the antibacterial activity of EOs should be indicated. What is the meaning of the letters a, b, c, d, e, f, g, h… above the bars? This should be indicated in the legend of figure 1. The letters are too small.

-In the legend of Figure 2, the meaning of 0.5, 1 and 0.5 ml should be explain in each case and the standards used.

-Figure 6 and 7 should be presented in a more clear way. For example bars for controls, ethanol solutions, Tw80 and Pickering solutions could be grouped.

- The meaning of O/W should be explained.

Reviewer 2 Report

General comment:

The manuscript  by Balázs et al studied the antibacterial activity of essential oils of cinnamon bark, clove,peppermint and thyme against Haemophilus influenzae and H.parainfluenzae  as well as the biofilm inhibition of different formulations of these EOs. The importance of essential oils has been increasing due to the high degree of antibiotic resistance.  The authors used O/W type Pickering nano-emulsions stabilized with silica nanoparticles from each oil to increase their water solubility and thin-layer chromatography-direct bioautography (TLC-DB) to evaluated the anti-Haemophilus activity of these emulsions. However, the methodology is incomplete  and the results are not clear and relevant informations are missing. The present paper could be acceptable for publication if the authors address the following comments.

Specific comments:

Introduction: The authors mentioned in the Introduction the impact of respiratory tract infections for public health. Did the authors tested other lung pathogens, such as Staphylococcus aureus or Pseudomonas aeruginosa? The authors commented that some of these essential oils have already been tested in other bacteria, but I think it would be a useful comparison if those representative bacteria were included in these ms. Considering the relevance of TLC-DB method for oil properties evaluation, the use of other species would also increase the reliability of the analysis.

Results: As already mentioned, the results are not well presented in this ms.

Section 2.2.1Lines 92-93 and Figure 1

Figure 1 shows the activity of the EOs without TLC separation. However, it is no clear how the authors did the experiment and obtain the results plotted in Figure 1. In Material and Methods, page 10, lines 314-322, the antibacterial effect of EOs without TLC separation is mentioned but the metodology used to get the results is not described.The authors only mentioned the activities  of the EOs components after TLC separation. A TLC-DB dot blot technique  or dot-blot bioautography is used for evaluation of a total activity of a whole sample (nonseparated sample). This tecnhique was employed by Jesionek et al., 2017 (Journal of liquid chromatography & related technologies, 40 (5-6): 292-296. http://dx.doi.org/10.1080/10826076.2017.1298031. Did the authors used a dot-blot tecnhique to get the results? So, the technique employed should be described in this Section and the Results inserted in the ms as an additional Figure.

Section 2.2.2 -Title

I think that the title could be changed to: Antibacterial activity of the main components of EOs by TLC-DB method.

Figures 2-5 have a unique legend. Too many informations and difficult to understand. What the authors think about separate Figure 2 from Figures 3,4 and 5 ? Figure 2 shows the results of EO from cinnamon bark and the solvent used to separate the components was different ( dichloromethane) from the solvent system used for peppermint, clove and thyme EOs (toluene-ethyl acetate 93:7 v/v). Figure 2C and 2D

Figure 2C and 2D- It is difficult to say that only the band with Rf= 0.35 has antimicrobial activity against both bacteria tested. Or the photo is not well defined?

Concerning the legend from Figures 2,3,4 and 5 we suggest that the authors could enumerate the standards from 1-10 and the numbers could be attributed according to their Rfs . For example: mix of standards: 1-menthol (Rf=0.31), 2-linalool (Rf=0.33), etc.And the standard eugenol has Rf=0.52 in toluene-ethylacetate 93:7 v/v and Rf= 0.76 in dichloromethane. It is important to add this information for a better understanding.Also in all TLC plates (A,B,C,D) the applied volumes of the EOs and the standards ( 0.5 and 1µl) could be included in the legend of the figure because they are the same. Lanes 1 and 2 could be the initials of the EO such as Cn, Cl, Pp and Th. Lane 3 corresponds to standards ( 1-10) and could be S. The number of the standards could stay as shown in the Figures.

Figure 3C and 3D- According to the text (lines 111-112), only the spot with Rf=0.52 identified as eugenol was active against both bacteria. And the spots with lower Rfs?

Figure 5C and 5D- It is clear that the spots corresponding to linalool (Rf=0.33) and thymol (Rf=0.56) present antibacterial activities.

Section 2.3- Preparation and characterization of Pickering nano-emulsion of EOs

This section is too theorethical and could be reduced. The authors could insert the paper of Horváth et al, 2018 (Flavour Fragr.1-12). The results are shown in Table 2. We think that this Section could start from line 125 to line 130. Some informations however could be included in this Section. A definition of conventional emulsion of essential oils. Are made with Tween 30? And ET in 20ET means ethyl group? 

Section 2.4- Anti-biofilm activity

Line 133- Why MIC/2 was used? Did the authors also test other concentrations, such as MIC values?

Line 134- Three different formulations of the EOs were tested.One of the formulation is the ethanolic solution.How did the authors prepare the ethanolic solution? Using 70% ethanol?

Lines 150-152- What does it mean?

Lines 141-144- The authors observed that Pickering nano-emulsions displayed the best results in bacterial biofilm formation.Did the authors check the citotoxicity of these preparations in order to see if the concentrations used are not toxic to mammalian cells?

Figures 6 and 7 are not well presented. The abscissa in the graphic (Figure 6) is too crowded and it is difficult to read . So, we suggest the substitution  of the initial of the EOs ( CP, CE, etc) to numbers ( 1, 2, etc). For example: (1) Control Pickering; (2) Control ethanol/ethanolic solution, etc. The same comments applied for Figure 7.

Table 1- Page 7 – Please change the title to : Average values of volatile compounds from EOs of Pepermint (1),Thyme (2), Clove (3) and Cinnamon(4) from three parallels experiments.

Round 2

Reviewer 2 Report

The revised ms was greatly improved. However a few corrections still need to be done.

Specific comments:

Section 2.2.1 –lines 92-93Reference # 22 is not related to the dot-blot bioautography. Please, delete this reference and keep only the reference # 23 from Jesionek et al., 2017. There is no Section 4.3.2.2 in the revised ms. And as cited by the authors, reference # 23 is not inserted in Section 4.3. Please, could the authors include this reference? Figures 2C and 2D – We agree. However, the sentence in lines 113-114 could be more clear such as: “α-terpineol (Rf=0.35) presente in the E) of cinammon showed activity against both bacteria”.

Also lines 117-118 could be :” Menthol (Rf=0.31) present in the peppermint oil and the standard inhibited the growth of bacteria” (Figure 4).

Line 142- We think that the correct is absolute etanol or ethanolic solution but not absolute ethanolic solution, because absolute etanol is not a solution. And what is the ethanolic solution concentration? 70%?

So, the revised ms could be acceptable for publication if the authors address this comments.

Author Response

Response for Reviewer 2:

1. We removed the reference #22, and we kept only the reference #23 (Jesionek et al. 2017) in the section 2.2.1. We renumerated the References, so the Jesionek et al. 2017 has the number 22 now in the references section. We also inserted this reference into 4.3 section. We renumbered the subsections and now there is 4.3.3. Post-Chromatographic Detection subsection in the manuscript.

2. We modified the sentence line 110-111: from „Moreover, α-terpineol showed activity at Rf= 0.35 in the EO of cinnamon.” to „Moreover, α-terpineol (Rf=0.35) in the EO of cinnamon showed activity against both bacteria.”

3. We modified the sentence line 114-115: from „Menthol in the peppermint oil, and of standard inhibited the growth of bacteria at Rf= 0.31.” to “Menthol (Rf=0.31) in the peppermint oil and the standard of menthol inhibited the growth of bacteria (Figure 2).

4. We specified the absolute ethanol expression in the whole manuscript. The word of ethanolic solution was changed to sample with absolute ethanol, and the word of emulgeant was changed to surfactant as well.